Ecomorphological inferences in early vertebrates: reconstructing Dunkleosteus terrelli (Arthrodira, Placodermi) caudal fin from palaeoecological data

Ferrón Humberto G. Humberto.Ferron@uv.es 1
Martínez-Pérez Carlos 1 2
Botella Héctor 1
1 Cavanilles Institute of Biodiversity and Evolutionary Biology, University of Valencia , Paterna , Spain
2 School of Earth Sciences, University of Bristol , Bristol , United Kingdom
Marsicano Claudia
Electronic publication date: 2017 Dec 6
Publication date: 2017
Volume: 5
Electronic Location ID: e4081
Received 2017 Jun 16; Accepted 2017 Nov 1
Copyright: ©2017 Ferrón et al.
Copyright year: 2017
Copyright holder: Ferrón et al.
License: This is an open access article distributed under the terms of the Creative Commons Attribution License, which permits unrestricted use, distribution, reproduction and adaptation in any medium and for any purpose provided that it is properly attributed. For attribution, the original author(s), title, publication source (PeerJ) and either DOI or URL of the article must be cited.
License URL: https://creativecommons.org/licenses/by/4.0/

Keywords: Ecomorphology, Early vertebrates, Sharks, Geometric morphometrics, Dunkleosteus terrelli, Caudal fin, Size, Palaeoart

Funding: European Regional Development Fund (ERDF) Spanish Ministry of Economy and Competitiveness CGL2014-52662-P Valencian Generality GV/2016/102 Spanish Ministry of Education, Culture and Sport FPU13/02660 This work was supported by the European Regional Development Fund (ERDF) ‘Una manera de hacer Europa’, the Spanish Ministry of Economy and Competitiveness (Research Project CGL2014-52662-P) and the Valencian Generality (Research Project GV/2016/102). Humberto G. Ferrón is a recipient of a FPU Fellowship from the Spanish Ministry of Education, Culture and Sport (Grant FPU13/02660). The funders had no role in study design, data collection and analysis, decision to publish, or preparation of the manuscript.

==============================
Our knowledge about the body morphology of many extinct early vertebrates is very limited, especially in regard to their post-thoracic region. The prompt disarticulation of the dermo-skeletal elements due to taphonomic processes and the lack of a well-ossified endoskeleton in a large number of groups hinder the preservation of complete specimens. Previous reconstructions of most early vertebrates known from partial remains have been wholly based on phylogenetically closely related taxa. However, body design of fishes is determined, to a large extent, by their swimming mode and feeding niche, making it possible to recognise different morphological traits that have evolved several times in non-closely related groups with similar lifestyles. Based on this well-known ecomorphological correlation, here we propose a useful comparative framework established on extant taxa for predicting some anatomical aspects in extinct aquatic vertebrates from palaeoecological data and vice versa. For this, we have assessed the relationship between the locomotory patterns and the morphological variability of the caudal region in extant sharks by means of geometric morphometrics and allometric regression analysis. Multivariate analyses reveal a strong morphological convergence in non-closely related shark species that share similar modes of life, enabling the characterization of the caudal fin morphology of different ecological subgroups. In addition, interspecific positive allometry, affecting mainly the caudal fin span, has been detected. This phenomenon seems to be stronger in sharks with more pelagic habits, supporting its role as a compensation mechanism for the loss of hydrodynamic lift associated with the increase in body size, as previously suggested for many other living and extinct aquatic vertebrates. The quantification of shape change per unit size in each ecological subgroup has allowed us to establish a basis for inferring not only qualitative aspects of the caudal fin morphology of extinct early vertebrates but also to predict absolute values of other variables such as the fin span or the hypocercal and heterocercal angles. The application of this ecomorphological approach to the specific case of Dunkleosteus terrelli has led to a new reconstruction of this emblematic placoderm. Our proposal suggests a caudal fin with a well-developed ventral lobe, narrow peduncle and wide span, in contrast to classical reconstructions founded on the phylogenetic proximity with much smaller placoderms known from complete specimens. Interestingly, this prediction gains support with the recent discovery of fin distal elements (ceratotrichia) in a well preserved D. terrelli, which suggests a possible greater morphological variability in placoderm caudal fins than previously thought.

Introduction

The fossil record provides a unique and fascinating glimpse into the lives of extinct creatures that once inhabited the Earth. Unfortunately, the palaeobiological information contained in the fossils is, in most cases, rather incomplete owing to different taphonomical processes (Allison & Bottjer, 2011 and references therein). In consequence, the understanding of numerous anatomical, ecological or evolutive aspects of extinct (but also living) organisms largely rely on the application of novel statistical procedures and specialized analytical methods that allow testing and compensating such preservational biases (Harper, 2008; Sepkoski & Ruse, 2009).

The presence or absence of mineralized tissues constitutes one of the main biasing factors in this sense, constraining in a considerable way the preservation potential of some groups or specific anatomical structures (e.g., Allison, 1986; Allison & Briggs, 1993; Briggs & Kear, 1993; Kidwell & Flessa, 1996; Briggs, 2003; Allison & Bottjer, 2011). Among vertebrates, skeleton mineralization occurred in the Cambrian at very early stages in the evolution of the group (Donoghue & Sansom, 2002), with the appearance of tooth-like structures in conodonts and dermo-skeletal elements, such as large bony plates or minute scales, in more derived forms (Sansom et al., 1992; Donoghue & Sansom, 2002; Donoghue, Sansom & Downs, 2006; Sire, Donoghue & Vickaryous, 2009; Murdock et al., 2013). However, large dermal plates are typically restricted to the cephalo-thoracic region in early vertebrate groups (e.g., placoderms and most agnathan ostracoderms), and in the vast majority of cases scales, that normally cover the whole body surface, disarticulate to isolated elements during preservation (Janvier, 1996). In consequence, although the fossil record of these mineralized structures is very abundant, they provide very little information about the whole external aspect of these animals. The late evolution of well-ossified endoskeletons in Silurian times (only present in osteichthyes) (Donoghue & Sansom, 2002; Donoghue, Sansom & Downs, 2006) just aggravates this problem, leading in conjunction to a very vague idea of the general body shape and size of a big number of Palaeozoic early vertebrate representatives, especially in regard to their post-thoracic region.

Previous reconstructions of most early vertebrates known from partial remains have been wholly based on phylogenetically closely related taxa (see some examples in Frickhinger, 1995; Janvier, 1996; Long, 2010; Cuny, 2013). However, the morphology of all biological structures is the result of a combination of several factors, including not only the phylogenetic legacy of the group but also morphogenetic, environmental and functional aspects (Seilacher, 1991). Interestingly, the shape of the body and the fins of living aquatic vertebrates is determined, to a large extent, by their swimming mode (e.g., Breder, 1926; Webb, 1975; Webb, 1988; Lindsey, 1978; Webb & Buffrénil, 1990; Thomson & Simanek, 1997; Sfakiotakis, Lane & Davies, 1999) and feeding niche (Webb, 1984). In fact, the functional constraints of different types of locomotion are so strong that different ecomorphotypes of fishes are clearly recognizable (e.g., Breder, 1926; Webb, 1975; Lindsey, 1978), having evolved several times in non-closely related groups adapted to similar lifestyles (Lindsey, 1978; Fletcher et al., 2014).

Thus, given the well-recognized relationship between the mode of life and body pattern in living aquatic vertebrates, we propose that the identification of ecomorphological relationships and the morphometric characterization of ecomorphotypes could be very useful for inferring the morphology of some missing structures in early vertebrate taxa. In this sense, living sharks can be considered as a suitable modern group for exploring this issue, showing an important ecological diversity and a wide range of body sizes. With this aim, we have assessed the relationship between the locomotory patterns and the morphological variability of the caudal region in extant sharks by means of geometric morphometrics and allometric regression analysis. We present a comparative framework for predicting some anatomical aspects in extinct species from palaeoecological data, and vice versa.

Dunkleosteus terrelli as a case of study

Dunkleosteus terrelli, a giant carnivorous placoderm that inhabited the seas of Euramerica during the Late Devonian (Carr, 2010; Carr & Jackson, 2010), constitutes one of the most representative examples in this regard. This large apex predator was equipped with sharp bladed jaws and one of the most rapid and powerful bites known in both living and extinct animals (Anderson & Westneat, 2007; Anderson & Westneat, 2009). Given its large size and fearsome appearance, D. terrelli has been the focus of interest for the general public during decades, becoming one of the most iconic fossils in the latter years. However, despite popular interest in the species D. terrelli is only known from disarticulated plates of the head shield, and a few articulated remains of incomplete pectoral fins (Carr, 2010; Carr, Lelièvre & Jackson, 2010). For this reason, its general body shape and size remain unknown and to date reconstructions of D. terrelli have relied on the morphology of smaller arthrodire placoderms known from complete specimens (e.g., Coccosteus in Heintz, 1932). As a consequence, some anatomical features of these taxa, such as the presence of tails with low heterocercal angles and poor developed ventral lobes and/or macruriform bodies, have been usually represented in publications and palaeoartistic reconstructions of D. terrelli (e.g., Heintz, 1932 fig. 90; Carr, 1995 fig. 17; Frickhinger, 1995 p. 136; Cuny, 2013 fig. 7). Here, based on the established ecomorphological framework, we propose a more parsimonious reconstruction of the caudal region of this flagship species from an ecological-proximity criterion, assuming analogy with living active pelagic sharks on the basis of morphological (Barron & Ettensohn, 1981), biomechanical (Anderson & Westneat, 2009), paleobiogeographic (Carr, 2009; Carr & Jackson, 2010) and taphonomic evidence (Carr, 2010; Carr & Jackson, 2010) (see ‘Discussion’ for a detailed assessment of the ecology of this taxon).

Figure 1 Descriptive diagrams showing (A–B) the landmark and wireframe configurations used in the geometric morphometric analyses of sharks and (C–D) the variables considered for the total body length estimations of Dunkleosteus terrelli.

Position of the landmarks (red points) and wireframe configurations (red dashed lines) chosen for the analyses of (A) the whole group of sharks and the ecological subgroups of demersal and squalomorph species; and (B) the ecological subgroup of active pelagic species. Shark drawings modified from Ebert, Fowler & Compagno (2013). Landmark 1, tip of the snout. Landmark 2, most posterior part of the eye. Landmark 3, uppermost part of the first gill opening. Landmark 4, pectoral fin origin. Landmark 5, lowest point on dorsal border of the caudal peduncle. Landmark 6, distal tip of the dorsal caudal-fin lobe. Landmark 7, distal tip of the ventral caudal-fin lobe. Landmark 8, highest point on ventral border of the caudal peduncle. Landmark 9, uppermost part of the fifth gill opening. Landmark 10, caudal fin posterior notch. Landmarks 1, 4–8 are type 2 and landmarks 2, 3 and 9 are type 1. (C) Upper Jaw Perimeter (UJP) measurement taken on the D. terrelli assembled specimens (CMNH 5768, CMNH 7424, CMNH 6090, CMNH 7054) and (D) Jaw Measurements (JMs) taken on the D. terrelli inferognathals. D. terrelli drawings modified from Carr & Jackson (2010). PN, postnasal plate; R, rostral plate; SO, suborbital plate; asterisks indicate the position of the quadratomandibular articulation.

Materials & Methods

Geometric morphometrics, PCA and regression analyses in sharks

Morphological variability of the caudal region of extant sharks was analysed in a selection of 31 species spanning the ecological, morphological and taxonomical diversity of the group (see Data S1). We have defined eight landmarks of type 1 and type 2 (Fig. 1A) that were digitized on lateral-view illustrations of sharks from Ebert, Fowler & Compagno (2013) by means of tpsDig1 software v.1.4 (Rohlf, 2004). The superimposition of landmark configurations was carried out with Generalized Procrustes Analysis (GPA) using MorphoJ software v. 1.06d (Klingenberg, 2011). This procedure allows the removal of variations in translation, rotation and size from the original landmark configurations. Procrustes coordinates were transformed into a covariance matrix and subjected to principal component analysis (PCA) also with MorphoJ software v. 1.06d (Klingenberg, 2011). The degree of homoplasy was checked by plotting the phylogeny of the studied species on the PCA morphospace and PCA results were interpreted categorizing sharks both into taxonomic groups (at order level) and according to their mode of life or locomotion capabilities (following the classification of Thomson & Simanek, 1997) in demersal, squalomorph, generalised and fast swimming pelagic sharks; the last two groups were reunified here into active pelagic sharks). Phylogenetic signal was checked in MorphoJ software v. 1.06d (Klingenberg, 2011) considering the phylogenetic relationships proposed by Vélez-Zuazo & Agnarsson (2011). Regression analyses between shape and total body length were performed for both the totality of sharks and each of the ecological subgroups in order to detect allometric changes in caudal fin morphology (two additional landmarks were considered for the analysis of active pelagic sharks, Fig. 1B). For that, a permutation test (number of permutations = 10,000) was carried out using tpsRegr v.1.4.1 (Rohlf, 2011) and MorphoJ software v. 1.06d (Klingenberg, 2011). The expected caudal fin morphology of D. terrelli was inferred from its total body length estimates (see below) by interpolation in the regression analysis of active pelagic sharks assuming ecological affinity. Finally, Pinocchio effect (which refers to the fact that variance seen at some specific landmarks is distributed across all landmarks during Procrustes Superimposition; Von Cramon-Taubadel, Frazier & Lahr, 2007) was tested comparing RFTRA (Resistant Fit Theta-Rho Analysis) and GPA superimpositions in IMP software CoordGen8 (Sheets, 2014). A validation test was performed by inferring the caudal fin morphology of several living shark species spanning the phylogeny of the group and covering a wide range of body sizes.

Total body length estimation of Dunkleosteus terrelli

Total body length of D. terrelli was estimated in a regression analysis between the total body length (TBL) and the upper jaw perimeter (UJP) of 245 extant sharks, belonging to 14 different species, from morphometric data compiled in Lowry et al. (2009) (Data S2). UJP was measured in four assembled (skull and thoracic armor) specimens of D. terrelli hosted in the Cleveland Museum (Ohio, USA) (CMNH 5768, CMNH 7424, CMNH 6090, CMNH 7054). Measurements were taken contouring the anterior margin of the suborbital, postnasal and rostral plates and considering quadratomandibular articulations as endpoints (Fig. 1C). However, the remains of the largest D. terrelli described up to date consist of an isolated partial inferognathal that only preserves the anterior half (CMNH 5936). As a consequence UJP cannot be directly measured on this specimen for body length estimations. For this reason, UJP of CMNH 5936 was approximated from additional regression analyses between the UJP and five inferognathal metric variables (JM1-5) measured on the four assembled specimens (Fig. 1D).

Results

Geometric morphometrics, PCA and regression analyses in sharks

Comparison between GPA and RFTRA superimpositions has allowed us to discard the existence of a Pinocchio effect in all the performed analyses (Data S3). Significant phylogenetic signal has been detected neither in the whole group of sharks (P-value: 0.18) nor in any ecological subgroup (P-value: 0.44, 0.31 and 0.48 for demersal, squalomorph and active pelagic sharks, respectively). PCA results for the whole group of sharks are shown in Fig. 2. PC1 explains 41.7% of the total variance whereas PC2 explains 32.7%. The observed morphological variability in caudal fins of sharks can be summarized in shape changes affecting mainly the relative fin span (represented by PC1 axis) and fin length (represented by PC2 axis). The mapped phylogeny and the distribution of taxonomic groups within the PCA graph reveal an important degree of morphological homoplasy in the caudal fin of species from different orders (Fig. 2A). Conversely, groups defined by Thomson & Simanek (1997), which reflect different modes of life, occupy more localized distributions within the PCA morphospace (Fig. 2B). Demersal sharks, restricted mainly to the highest values in PC1, are characterized by caudal fins with low spans; squalomorph sharks, occupying intermediate values in PC1 and high values in PC2, are characterized by comparatively short caudal fins; and, finally, active pelagic sharks, situated in the lowest values of PC1, are characterized by caudal fins with wide spans. Positive allometry affecting mainly caudal fin span has been detected in the whole group of sharks (P-value <0.001), squalomorphs (P-value: 0.097, significant at alpha = 0.1) and active pelagic sharks (P-value: 0.009), whereas no allometric changes have been found in demersal sharks (P-value: 0.380) (Fig. 3). Validation analysis shows that the established methodological framework is able to properly predict the caudal fin morphology of the vast majority of tested sharks (confined within the individual confidence interval boundaries) (Fig. S1).

Figure 2 Principal Component Analysis (PCA) results for the whole group of sharks.

PCA plots of the first two PC axes showing the distribution of (A) taxonomic groups at order level and (B) modes of life according to Thomson & Simanek (1997). The phylogenetic tree (modified from Vélez-Zuazo & Agnarsson, 2011) is mapped into the PCA morphospace. Wireframe configurations show shape changes from the negative to the positive extreme of the axes (black and red respectively).

Figure 3 Allometric regression analysis results for (A) the whole group of sharks and the ecological subgroups of (B) demersal, (C) squalomorph and (D) active pelagic shark species.

Wireframe configurations show shape changes from the negative to the positive extreme of the axis (black and red respectively). Upper and lower limits of 95% mean confidence intervals are showed with dashed lines for each regression analysis.

Total body length estimation and caudal fin reconstruction of Dunkleosteus terrelli

Values for UJP and JMs of D. terrelli studied specimens are given in Table 1. Regression analysis between both variables shows a good fit to a linear model (R2 above 0.9 in all cases; Table 2). Estimates of UJP for CMNH 5936 specimen ranged between 118.2 cm and 163.8 cm (Table 2), but 155.7 cm has been considered as the most accurate estimate, obtained from the best-fitted regression model, and has been used for subsequent steps. Regression analysis between UJP and TBL raw data of extant sharks also shows a good fit to a linear model (R2 = 0.86, TBL (cm) = 5.33*UJP (cm) + 48.66). Total body lengths estimations of 3.20, 5.34, 5.76, 6.88 and 8.79 m were obtained for D. terrelli specimens CMNH 5768, CMNH 7424, CMNH 6090, CMNH 7054 and CMNH 5936 respectively.

Table 1 Upper Jaw Perimeter (UJP) and Jaw Measurements (JM) of Dunkleosteus terrelli assembled specimens.

Specimen	UJP (cm)	JM1 (cm)	JM2 (cm)	JM3 (cm)	JM4 (cm)	JM5 (cm)	
CMNH 5768	120.0	14.0	9.5	17.0	15.0	27.0	
CMNH 7424	51.0	6.5	4.5	8.5	7.0	13.3	
CMNH 6090	91.0	11.0	7.0	11.5	12.0	20.0	
CMNH 7054	99.0	10.0	8.5	13.5	14.0	23.0	

Predicted caudal fin morphologies for each of the D. terrelli specimens show well-developed ventral lobes, narrow peduncles, wide spans and high heterocercal and hypocercal angles (ranging from 22° to 30° and from 44° to 47° respectively) (Fig. 4A). Reconstruction of the largest specimen (8.79 m) is shown in Fig. 4B, with the highest caudal fin heterocercal and hipocercal angles (30° and 47° respectively), the best-developed ventral lobe (1.26 m) and the widest fin span.

Figure 4 Caudal fin shape inferences in Dunkleosteus terrelli.

(A) Predicted caudal fin shape of each specimen of D. terrelli (in black), showing the upper and lower 90% individual confidence interval boundaries (in red and blue respectively). (B) Palaeoartistic reconstruction of a 8.79 meters D. terrelli (courtesy of Hugo Salais, HS Scientific Illustration).

Table 2 Regression results between Upper Jaw Perimeter (UJP) and Jaw Measurements (JM) of Dunkleosteus terrelli assembled specimens and inferred values of UJP for the inferognathal specimen CMNH 5936.

Regression	Equation	R2	CMNH 5936 UJP (cm)	
UJP * JM1	y = 8.963x − 2.7407	0.9208	118.3	
UJP * JM2	y = 13.084x − 6.2423	0.9704	163.8	
UJP * JM3	y = 7.8265x − 8.5597	0.9346	140.1	
UJP * JM4	y = 7.9737x − 5.4342	0.9653	146.1	
UJP * JM5	y = 4.9664x − 13.176	0.9872	155.7	

Discussion

Ecomorphological traits of shark caudal fin

Caudal fin morphology and locomotion capabilities of sharks

Body design of aquatic vertebrates is closely related with swimming mode (e.g., Breder, 1926; Webb, 1975; Webb, 1988; Lindsey, 1978; Webb & Buffrénil, 1990; Thomson & Simanek, 1997; Sfakiotakis, Lane & Davies, 1999) and feeding niche (Webb, 1984). This morphological convergence is especially noticeable in the shape and size of the body and caudal fin of the groups where the tail is the main structure involved in thrust generation (body-caudal fin propulsion) (Lindsey, 1978). Here, the study of the caudal fin morphological variation of 31 different species of sharks provides additional evidence in this sense, showing a strong morphological convergence in non-closely related shark species that share similar modes of life (compare Figs. 2A and 2B). The shark body patterns proposed by Thomson & Simanek (1997) are recognizable after our morphometric analysis as extremes of a morphological continuum which is probably due to a graduation in locomotion capabilities (Fig. 2B). On the other hand, some specific morphological features, such as thunniform body and/or caudal fins with high aspect ratio, have evolved independently in both living and extinct groups of aquatic vertebrates adapted to strong continuous swimming (Motani, 2002; Donley et al., 2004; Lindgren et al., 2010; Lindgren, Kaddumi & Polcyn, 2013). Accordingly, our results support a notable homogeneity in the caudal fin morphology of active pelagic sharks, most of them sharing the possession of a well-developed ventral lobe and wide caudal fin span (Fig. 2B) that maximizes thrust and minimizes drag and recoil energy losses (Langerhans & Reznick, 2010).

Caudal fin allometry of sharks

Besides ecology, this study suggests that caudal fin morphology of sharks is also influenced by body size. We have found interspecific positive allometry affecting mainly the caudal fin span (Fig. 3A). This same phenomenon has been previously reported during the ontogeny of some phylogenetically distant aquatic vertebrates, including chondrichthyans (Fu et al., 2016), osteichthyans (Arata, 1954; Nakamura, 1985), cetaceans (Lingham-Soliar, 2005) and even fossil groups such as ichthyosaurs (Motani, 2002). The presence of this particular trend in so disparate groups responds to common physiological constraints. There is evidence that increased size results in lower muscle contraction frequency (Altringham & Johnston, 1990; Altringham & Young, 1991), which results in lower tail beat frequency (Wardle, 1975). Consequently, a loss in swimming speed and hydrodynamic lift is compensated with proportionally wider caudal fin spans (Motani, 2002; Lingham-Soliar, 2005). As expected, we have detected this phenomenon only in sharks with comparatively good swimming capabilities (i.e., squalomorph and active pelagic species, Figs. 3C and 3D) whereas it is absent in demersal species where hydrodynamic lift is not as important, with decreased need to control their position in the water column (Fig. 3B). These findings imply further evidence supporting the involvement of the caudal fin allometry found in many aquatic vertebrate groups as a compensation mechanism for the loss of hydrodynamic lift associated with the increase in body size.

Scope and limitations of the framework

Our work is the first to establish a comparative framework with geometric morphometrics of body-caudal fin propulsion on which to base ecomorphological inferences of extinct early vertebrates with body-caudal fin propulsion. The high predictive power of this methodology is demonstrated by the validation analysis, being able to properly infer the caudal fin morphology of an important number of the examined species (Fig. S1). We propose that this approach is applicable for both: (1) predicting some missing morphological information of the caudal fin region based on palaeoecological data, and (2) inferring swimming capabilities or some other ecological traits in species known from well-preserved complete specimens. The range of extinct taxa that could be potentially included in this model encompasses an important number of early vertebrates, including different groups of gnathostomes (e.g., many extinct groups of chondrychthyans, placoderms and ‘acanthodians’) and presumably a few agnathans with epicercal tails (e.g., osteostracans) (Janvier, 1996). However, the influence of structures that contribute to bouyancy (i.e., liver, swim bladders or lungs) on the allometric scaling factor of caudal fin spans should be quantified in future studies in order to improve the methodology here proposed and increase the accuracy of the derived predictions. Until then, this proposal can be considered as a suitable first-attempt model for supporting ecomorphological inferences of extinct early vertebrates.

Dunkleosteus terrelli as a case study

The lack of preserved post-thoracic remains of large pelagic arthrodires and the small size of the species known from complete specimens (see Denison, 1978) make it difficult to base ecomorphological inferences of D. terrelli on other placoderms. We propose that living sharks can be considered as suitable models for predicting characters strongly correlated with the lifestyle in placoderms, in cases like this, where no other closer related taxa with similar ecology are known. As in modern sharks, arthrodire placoderms were also body-caudal fin propulsors, generating thrust with lateral movements of their tails (Denison, 1978). Consequently, caudal fin morphology of D. terrelli, as the main propulsor organ, should also be subject to strong selection pressures driven by ecology, locomotion and body size.

Ecology of Dunkleosteus terrelli

Both the lifestyle and trophic position of this species are relatively well known regardless of the virtual absence of post-thoracic remains. D. terrelli has been interpreted as a placoderm with good swimming capabilities, probably being able to actively pursue its prey (Heintz, 1932; Carr, 1995). Recently, Carr (2010) has provided strong statistical evidence for active pelagic cruising in this species, based on taphonomical and sedimentological data. Furthermore, palaeogeographical data is in accordance with such inferences. Fossil remains of D. terrelli have been reported from the Upper Devonian of both the Appalachian Basin (EEUU) and the Tafilalt Basin (Morocco), suggesting a Euramerican distribution and supporting that this species was able to disperse throughout the Rheic Ocean, an ancient deep water biogeographic barrier (Carr, 2009; Carr & Jackson, 2010). Classically, anguilliform swimming mode, which implies large amplitude undulations of the whole body (Lindsey, 1978), has been presupposed for small elongated species and extrapolated for the entire group of placoderms (Stensiö, 1963; Thomson, 1971). However, more recently, some authors called into question this assumption proposing that other swimming modes could also be present within placoderms (Carr, 1995; Carr, Lelièvre & Jackson, 2010). In fact, anguilliform propulsion seems unlikely for most representatives of this group as the presence of the cephalic shield limits lateral movements of the head and part of the trunk (Carr, 1995). In consequence, other more efficient swimming modes such as subcarangiform or even thunniform, where undulations are restricted to the posterior part of the body (Lindsey, 1978), could be more plausible for at least some pelagic placoderms.

On the other hand, regarding trophic position, big arthrodires were situated at the top of the Late Devonian marine trophic pyramids constituting the earliest case of apex predatory vertebrates (Anderson & Westneat, 2009; Lamsdell & Braddy, 2009). The presence of powerful jaws, extremely fast bite and suction feeding mechanisms suggests predation on evasive, free-swimming and armoured animals, for example arthropods, ammonoids or other placoderms (Anderson & Westneat, 2009). In addition, some other anatomical evidence such as the terminal position of the mouth, the lateral compression of the body and the reduction of the thoracic shield and pectoral spines also support a predatory lifestyle in the water column (Barron & Ettensohn, 1981 and references therein). Besides all these indirect inferences, teeth of a small chondrichthyan (i.e., Orodus) have been discovered in association with D. terrelli remains and regarded as putative stomach contents (Carr & Jackson, 2010), thus constituting a possible direct evidence of the diet of this massive placoderm (but see Williams, 1990). Interestingly, Orodus spp. have been interpreted as tachypelagic chondrichthyans (i.e., high-speed pelagic species) according to the general morphology of some complete specimens from the Pennsylvanian (Carboniferous) (Compagno, 1990) and the arrangement and morphology of their dermal denticles, comparable to those of the scales of living fast pelagic shark (Raschi & Musick, 1986). In summary, taken altogether, the evidence support that D. terrelli was a big cruiser with good swimming capabilities, situated at the top of the trophic pyramid as active predator. In consequence, both the ecology and trophic position of this species, and probably other big arthrodires, are comparable to those of living active pelagic sharks, being possible to establish an ecological analogy between both groups.

Body size of Dunkleosteus terrelli

Previous references to the size of D. terrelli are not based on quantitative approaches (Denison, 1978; Frickhinger, 1995; Anderson & Westneat, 2007; Anderson & Westneat, 2009; Albert, Johnson & Knouft, 2009; Carr, 2010; Long, 2010) and some of them are probably overestimates (e.g., 10 m in Anderson & Westneat, 2009). Size estimations of other big placoderms have been founded on corporal proportions of Coccosteus, extrapolating the ratio between some shield measurements and total body length calculated in complete specimens of this species (e.g., Gross, 1960; Young, 2005; Vaškaninová & Kraft, 2014). However, the reliability of such approximations should be questioned since the obtained estimates are too far from the usual range of lengths of Coccosteus cuspidatus (one order of magnitude higher) and the presence of allometry has been documented many times in some shield plates of different species (e.g., Werdelin & Long, 1986; Zhu & Janvier, 1996; Trinajstic & McNamara, 1999; Trinajstic & Hazelton, 2007; Olive et al., 2014), including C. cuspidatus itself (Miles & Westoll, 1968). Therefore, basing total body length estimations of D. terrelli on other placoderms could be again inadequate and different approaches should be applied. In this sense, some previous works have used extant sharks as useful models for estimating other body variables in placoderms (see (Carr, 2010)). Here, we have approximated the size of D. terrelli using the relationship between the total body length and the upper jaw perimeter established from several species of extant big pelagic sharks. A total body length of 8.79 m has been inferred for the biggest specimen of D. terrelli (CMNH 5936), being considerably larger than some of the previous more conservative estimates (e.g., between 4 and 5 m in Long, 2010; 4.6 m in Carr, 2010; and 6 m in Denison, 1978; Anderson & Westneat, 2007). The use of this approach, or other similar ones, for inferring body sizes of big placoderms can be preferable to the previous ones for two reasons: (1) the range of sizes of some living sharks is within the order of magnitude of sizes expected for D. terrelli, and (2) the mouth perimeter and body size are indirectly linked in marine predators by some ecological aspects such as the trophic level and prey size (e.g., Wainwright & Richard, 1995; Scharf, Juanes & Rountree, 2000; Karpouzi & Stergiou, 2003). Therefore, mouth size or equivalent variables such as the upper jaw perimeter can be suitable predictors of body size when considering fishes that share similar feeding niches, as big predatory pelagic sharks and D. terrelli presumably do.

Reconstruction of Dunkleosteus terrelli caudal fin

Previous reconstructions of D. terrelli based on the anatomy of the smaller arthrodire placoderm Coccosteus (e.g., Heintz, 1932) could be inaccurate as this implies the comparison of taxa coming from too different facies and taxonomical assemblages and, ultimately, with possibly disparate lifestyles (Carr, 2010). In fact, Miles & Westoll (1968) suggested that Coccosteus, although being well adapted to free swimming, would display also demersal habits using the bottom as an ambush site. Here, according to the pre-existing palaeoecological data and body size estimations of D. terrelli we propose a completely different reconstruction. Assuming ecological similarity with big living active pelagic sharks, our model predicts a caudal fin with a well-developed ventral lobe, high heterocercal and hypocercal angles, narrow peduncle and wide span (Fig. 4A). Interestingly, given the big size of this taxon, these inferences do not differ importantly if considering other less likely alternative scenarios as a slow-swimming benthopelagic lifestyle (i.e., assuming ecological analogy with squalomorph sharks (Fig. S2). This prediction gains support with the recent discovery of ceratotrichia in a pectoral fin of D. terrelli (Carr, Lelièvre & Jackson, 2010). This finding implies that fins of placoderms could exceed the limits thought up to date, confined to the extension of the fin basal or radial elements, thus offering a greater range of morphological variability and locomotor capabilities than previously suggested (Carr, Lelièvre & Jackson, 2010; Carr & Jackson, 2010). In addition to caudal fin morphology, some other aspects have been carefully considered with the aim of providing a fairly accurate whole reconstruction of D. terrelli in accordance with the existing scientific evidence (Fig. 4B). The shape and arrangement of the dermal plates as well as the position and extension of the sensory lines of the cephalic shield are based on the study of the assembled specimen CMNH 6090. Pectoral fin proportions follow the proposal of Carr, Lelièvre & Jackson (2010), supported by observations taken on partial articulated remains of this species. On the other hand, just a single dorsal fin is represented in agreement to the condition shown by all the other arthoridire placoderms where this structure is preserved (Denison, 1978). In this case, a phylogenetical-proximity criterion is followed as the number of dorsal fins usually remains constant at order level both in living and extinct fishes (Nelson, Grande & Wilson, 2016). Finally, countershading coloration and fusiform body, rather congruent with an active pelagic lifestyle, has been illustrated in contraposition to more eye-catching and macruriform representations with the broadest part of the body immediately behind the head (e.g., Heintz, 1932 fig. 90; Frickhinger, 1995 p. 136; Cuny, 2013 fig. 7).

Comments on the importance of ecomorphology on palaeoartistic reconstructions

Palaeoart plays an essential role popularising palaeontology, generating a visual idea for the general public of how extinct organisms were in life (Witton, Naish & Conway, 2014). In this sense, the use of rigorous methodologies and updated scientific information is essential for the production of accurate reconstructions and the appropriate scientific dissemination of the most current knowledge in palaeobiology. Otherwise, imprecise representations can have very negative effects, leading to misperceptions about extinct species that could become deeply rooted in the general public, especially in the case of taxa with great media impact (e.g., Glut & Brett-Surman, 1997). Several procedures have been proposed in order to carry out realistic reconstructions supported by the available scientific knowledge (Paul, 1987; Paul & Chase, 1989; Witmer, 1995; Sellers et al., 2009; Ghilardi & Ribeiro, 2010). Among them, phylogenetic bracket approaches have been commonly applied for reconstructing non-preserved soft tissues and other palaeobiological aspects on the basis of maximum likelihood criterion by comparison with phylogenetically closely related taxa (e.g., Witmer, 1995). This kind of approach can be really useful for reconstructing characters with strong phylogenetic load that remain constant within the taxonomic group regardless of other factors. However, the application of such methodologies in a loose way, in order to make inferences about morphological characters without an osteological control or other palaeobiological aspects such as behaviour, physiology or ecology, requires a high level of speculation and is extremely sensitive to convergence. In those cases, the application of other approaches that imply a “briefing” (e.g., Ghilardi & Ribeiro, 2010), aiming to integrate evidence from a broader spectrum of disciplines, could be more appropriate. In this sense, this work highlights the need of considering extant ecological analogues and palaeoecological data coming from biomechanical, taphonomical, sedimentological and palaeobiogeographical studies for helping us to know what some fossil early vertebrates looked like. Only in that way, their reconstructions will be in agreement not only with their phylogenetic legacy but also with the ecomorphological patterns related to their particular lifestyles, waiting for the discovery of new remains that shed light on their real appearance.

Supplemental Information

Data S1 Shark species included in the geometric morphometric analyses

Click here for additional data file.

Data S2 Total Body Length (TBL) and Upper Jaw Perimeter (UJP) data of extant sharks considered in this study. Data taken from Lowry et al. (2009)

Click here for additional data file.

Data S3 GPA (left) and RFTRA (right) superimpositions for the whole group of sharks and for each ecological subgroup. Points denote landmarks and crosses denote landmark centroids

Click here for additional data file.

Figure S1 Validation test results. Inferred caudal fin morphology (black dashed lines) of (A–G) squalomorph and (H–M) active pelagic living sharks, showing the upper and lower 90% individual confidence interval boundaries (red and blue dashed lines respectively)

(A) Echinorhinus brucus 310 cm, (B) Squalus cubensis 110 cm, (C) Centrophorus atromarginatus 94 cm, (D) Centroscyllium ritteri 43 cm, (E) Centroscymnus coelolepis 122 cm, (F) Somniosus microcephalus 730 cm, (G) Dalatias licha 182 cm, (H) Rhizoprionodon terraenovae 110 cm, (I) Sphyrna corona 92 cm, (J) Isurus paucus 430 cm, (K) Cetorhinus maximus 1,000 cm, (L) Rhincodon typus 2100 cm, (M) Carcharhinus longimanus 395 cm. Shark outlines modified from Ebert, Fowler & Compagno (2013).

Click here for additional data file.

Figure S2 Caudal fin shape inferences in Dunkleosteus terrelli assuming ecological analogy with squalomorph sharks

(A) Predicted caudal fin shape of each specimen of D. terrelli (in black), showing the upper and lower 90% individual confidence interval boundaries (in red and blue respectively). (B) Palaeoartistic reconstruction of a 7.33 meters D. terrelli (courtesy of Dr. Hugo Salais, HS Scientific Illustration).

Click here for additional data file.

We would like to thank the palaeoilustrator Mr. Hugo Saláis (University of Valencia) for his artistic contribution in Fig. 4; the Dr. Vélez-Zuazo (University of Puerto Rico) for making available to us the phylogeny (NEXUS) files of Vélez-Zuazo & Agnarsson (2011); and the Dr. Amanda McGee and Dr. Michael Ryan (Cleveland Museum) for providing to us the measurements of Dunkleosteus terrelli specimens. We are also thankful to Dr. Soledad De Esteban Trivigno (Institut Català de Paleontologia) and Borja Figueirido Castillo (Universidad de Málaga) for their comments on early versions of the manuscript and advice on the methodological approach. We acknowledge the comments of Dr. Claudia Marsicano (Academic Editor), Dr. Robert Carr and Dr. Thomas Fletcher (referees) and an anonymous reviewer that have considerably improved the final manuscript.

Additional Information and Declarations

Competing Interests

Author Contributions

Data Availability

The authors declare there are no competing interests.

Humberto G. Ferrón conceived and designed the experiments, performed the experiments, analyzed the data, contributed reagents/materials/analysis tools, wrote the paper, prepared figures and/or tables.

Carlos Martínez-Pérez conceived and designed the experiments, contributed reagents/materials/analysis tools.

Héctor Botella wrote the paper, reviewed drafts of the paper.

The following information was supplied regarding data availability:

The raw data has been provided as Supplemental Files.

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
