# Peer review of "Ecomorphological inferences in early vertebrates: reconstructing Dunkleosteus terrelli (Arthrodira, Placodermi) caudal fin from palaeoecological data"

_PeerJ, doi:10.7717/peerj.4081_

## Round 0.1 · original submission · Major Revisions

Dear Mr Ferron,

Your paper "Ecomorphological inferences in early vertebrates: reconstructing Dunkleosteus terrelli (Arthrodira, Placodermi) from palaeoecological data", co-authored with Martínez-Perez and Botella, has now been reviewed by three referees, and all agree that it deserves publication. Nevertheless, two of them support publication after major revision, an opinion that I support as Editor.

Reviewer #2 and reviewer #3 concur that the Ms is very interesting but several major issues need to be better supported. It is crucial, as was pointed out by Reviewer #2, that the rationale behind your conclusions perceives not arbitrary as, for example, the election of the "great white" shark as a suitable model to reconstruct Dunkleosteus. In this context, I strongly recommend you a fuller comparison between modern sharks and Dunkleosteus in order to determine which shark ecological group is the ideal analogue and thus the chain of inferences that leads you to the proposed life-mode reconstruction. I consider it is very important the discussion of alternative interpretations/scenarios, if only to rule them out.

Please, also pay particular attention to the detailed comments in your Ms related to the language made by reviewer #1 and reviewer #2.

So, I am requesting that you revise the parts mentioned above, simply to make your general discussion and conclusions better supported. As the revisions required are extensive enough, another round of review may be necessary when you resubmit your revised manuscript.

Thank you for submitting your Ms to PeerJ and I look forward to receiving your revision.

Sincerely,
Claudia Marsicano

·

Basic reporting

refer to general comments for the author

Experimental design

no comment

Validity of the findings

refer to general comments

Additional comments

I recommend publication with minor revisions as listed below.

I am not familiar with the software programs used so I will not comment on their efficacy. I will list below comments and errors by their line number.

Ln 45 hypocercal instead of hipo-

Ln 49 founded instead of funded

Ln 72 pteraspidomorphs instead of pteraspidomorfs

Ln 103 Add Carr (2010) to the references about Dunkleosteus inhabiting the seas of Euramerica.

Ln 110 Add Carr (2010) to the references, since it directly address the pattern of disarticulation in Dunkleosteus.

Ln 112 In line 333 you refer to Coccosteus as a “close related placoderm.” In this line you refer to Coccosteus as a ”smaller basal arthrodire.” Neither case is accurate. Coccosteus is a member of the coccosteomorph arthrodires while Dunkleosteus is a member of the pachyosteomorph arthrodires. These two groups are sister groups, thus neither is more basal to the other. Terminology more appropriate might be what you used to describe sharks as “non-closely related” since the two arthrodires are from distinct orders.

Ln 124 In this line you refer to type 1 and type 2 and direct the reader to Fig. 1A where you show the 8 landmarks. Is Fig. 1A the type 1 and Fig. 1B the type 2? You may want to note then that type 2 has two additional landmarks as noted in Fig. 1B. You could add this within the parentheses. Also, in the figure caption you should identify which figure represents type 1 and type 2 so that the text and caption parallel each other.

Ln 142 Add “v.1.4.1” to the software (tpsRegr v.1.4.1) to be consistent with your references to other software packages (e.g., tpsDig1 software v.1.4).

Ln 149 Elsewhere in the text you only refer to TBL, not BL (e.g., in line 188 and Supplemental Data S2).

Ln 153-154 The plate names will not be familiar to non-placoderm workers so it would help if you labeled those plates on your Fig. 1C. As in Carr & Jackson, 2010: fig. 5-7, you can add SO, PN, and R to your Fig. 1C. For the site of the quadratomandibular articulation you can place an asterisk on the PSO plate. Be sure to define your abbreviations in the figure caption.

Ln 158 Since you use the abbreviation “JM” later (e.g., line 183) you should introduce it in this line, e.g., “five infergnathal metric variables (JM1-5).”

Ln 164-164 This sentence contains a double negative (No, with neither, nor). Either use “no, either , or” or possibly “Significant phylogenetic signals have been detected neither…nor….”

Ln 165 Add a comma after sharks (“sharks, respectively).

Ln 166-167 Drop the article “the” before the percentages, i.e., “PC1 explains 41.7% of the total variance whereas PC2 explains 32.7%.”

Ln 193 In this line you refer to “deep peduncles” while in line 341 you call it a “narrow peduncle.” Deep (to me) implies a wide peduncle so change line 193 to match line 341. Also in this line change hipocercal to hypocercal.

Ln 202 In the references, you list this as Thomson & Simanek, 1977.

Ln 212 Thunniform instead of tuniform (the same change elsewhere).

Ln 217 lobe instead of love

Ln 237 Change phrasing: I suggest “our work has allowed the establishment of a comparative framework”

Ln 246 bouyancy instead of bouayancy

Ln 258 Change phrasing: “As in sharks”

Ln 307 founded instead of funded

Ln 280 thunniforn instead of tuniform

Ln 286 mechanisms instead of mechanism

Ln 294 Is your reference to the entire book by Boucot or more specifically to Williams, 1990, Feeding behavior in Cleveland Shale fishes, p. 273-287. In Boucot, Evolutionary Paleobiology of Behavior and Coevolution. I think the specific reference would aid the reader in finding the reference.

Ln 307 founded instead of funded

Ln 325 Spell out the first use of the genus name in C. carcharias.

Ln 333 refer to line 112 comment. Does membership in a separate but sister order make it “close related”?

Ln 341 refer to line 193 comment

Ln 343 Instead of “involves” would “implies” be more appropriate?

Ln 358 Usually references to a figure in another document use a lower case “f” while references to figures in the manuscript use an upper case “F”. What are the instructions to authors guidelines for PeerJ?

Ln 369-370 In line 513 you refer to Paul, Chase & Hodges, 1989 as Paul & Chase and note that it is in book edited by Hodges. Which is correct?

Ln 374 Use either “This kind of approach can be” or “These kinds of approaches can be”

Ln 384 Phrasing: “how” versus “what”?

Ln 384-388 This is a fragmented sentence. Can you break it up into two or more sentences? It is hard to follow as it is now.

I checked that all the references are listed in the text and visa versa.

Figure 1 As noted above, it would help the non-placoderm reader to add plate abbreviations for the SO, PN, and R as well as a possible asterisk for the site of the jaw articulation on the PSO (refer to Carr & Jackson, 2010:fig. 5-7). Again, my question of adding type 1 and 2 in the figure caption to match their mention in the text.

Tables 1&2 (JM1-5) instead of simply (JM)

·

Basic reporting

In the interest of open and honest dialogue in science I do not wish to remain anonymous, and encourage the authors to contact me if they have any queries. Regards and best, Tom Fletcher, University of Leicester

• Clear, unambiguous, professional English language used throughout.
Suggested improvements:

L28 - “However, body design of fishes is determined in a large extent …” should be “determined to a large extent”.
L29 - “… being possible to clearly recognize different morphological traits that have evolved several times in non-closely related groups with similar lifestyles” I would suggest should be “making it possible to recognise”. I feel clearly recognising is too general in that instance, as it is only very extreme adaptations that are obviously for a specific purpose.

L45 - “fin span or the hipocercal” should be hypocercal

L46 - “The application of this ecomorphological approach to the concrete case of Dunkleosteus terrelli has led to a new reconstruction of this emblematic placoderm.” It is unclear what you mean here, what is a concrete case?
L49 - “in contrast to classical reconstructions funded on the phylogenetic proximity with much smaller placoderms known from complete specimens” Funded is an inappropriate word here. I would suggest “… reconstructions based on the…”

L52 - “which entails that the caudal fins of placoderms could have displayed a greater range of morphological variability than thought up to date”. I would suggest “which suggests a greater morphological variability in placoderm caudal fins than previously thought.”
L55 – “The Fossil Record…” should not be capitalised, suggest “The fossil record…”
L55 – “…glimpse into the extinct creatures” is unclear, suggest “glimpse in to the lives of extinct creatures”.
L70 – Sentence a little long “However, large dermal plates are typically restricted to the cephalo-thoracic region in early vertebrate groups (e.g., placoderms and most agnathan ostracoderms including pteraspidomorphs, galeaspids, osteostracans and pituriaspids) and scales, that normally cover whole body surface, easily disarticulate being found as isolated elements in the vast majority of cases (Janvier, 1996).” I’d suggest “However, large dermal plates are typically restricted to the cephalo-thoracic region in early vertebrate groups (e.g., placoderms and most agnathan ostracoderms), and in the vast majority of cases scales disarticulate to isolated elements during preservation (Janvier, 1996).”
L76 – Some repetition here, and a little unclear suggest deleting “ leading in conjunction to a very vague 79 idea of the general body shape and size of a big number of early vertebrate representatives, 80 especially in regard to their post-thoracic region.” Instead explain briefly (7-8 words) that endochondral bone has a greater preservation potential (with reference).
L85 – ‘Fabricational’ is not an appropriate word here, do you mean ‘developmental’? Nevertheless I would suggest “including not only the phylogenetic legacy of the species but also environmental and functional aspects (Seilacher, 1991).” I would use ‘species’ instead of ‘group’ here, since you are arguing (and I agree) that every species is uniquely shaped by the factors you list.
L86 – I suggest “… aquatic vertebrates is determined to a large extent by their swimming mode”
L88 – Suggest “… the functional constraints of different types of locomotion are so strong that…”
L93 – Suggest “…between the mode of life and body pattern in living aquatic vertebrates, we propose that the identification of ecomorphological relationships…”.
L96 – This is a very long sentence. Suggest “With this aim, we have assessed the relationship between the locomotory patterns and the morphological variability of the caudal region in extant sharks by means of geometric morphometrics and allometric regression analysis. We present a comparative framework for predicting some anatomical aspects in extinct species from palaeoecological data, and vice versa”
L103 – ‘Paradigmatic’ is not an appropriate word here. Suggest “Dunkleosteus terrelli is an exceptionally large and iconic carnivorous placoderm that inhabited the seas of Euramerica during the Late Devonian (Carr & Jackson, 2010).”
L106 – I suggest “Given its large size* and fearsome appearance, D. terrelli has been a focus of interest for the general public for decades.“. *I would also include a length in metres here, with reference.
L108 – Suggest “However, despite popular interest in the species D. terrelli is only known from disarticulated plates of the head shield, and a few articulated remains of incomplete pectoral fins (Carr, Lelièvre & Jackson, 2010).”
L111 – Suggest “ … remain unknown, and to date reconstructions of D. terrelli have relied on the morphology of smaller basal …”
L113 – I think the paragraph starting “As a consequence …” repeats the previous paragraph and I’d delete it
L128 – Suggest “This procedure allows the removal of variations in….”
L131 – Suggest “The degree of homplasy was checked by plotting … “
L163 – Should be “… of a Pinocchio effect …t”
L164 – Double negative, suggest “… detected either for the … “
L166 – Should be “…PC1 explains 41.7% of the total variance whereas PC2 explains 32.7%.”
L168 – Should be “… caudal fins of sharks …”
L184 – Should be “ … both variables shows a good fit to a linear model…”
L208 – This whole sentence is unclear and I think repeats what follows. If you want to keep this I’d suggest “The shark body patterns proposed by Thomson & Simanek (1997) are recognizable after our morphometric analysis. Separate groups, as extremes of a morphological continuum, are probably due to different locomotary styles (Fig. 2B).”
L214 – Should be “… adapted to strong continuous swimmimg… “
L217 – Should be “well-developed ventral lobe and wide …”
L224 – Suggest “There is evidence that increased size results in lower muscle contraction frequency (Altringham & Johnston, 1990; Altringham & Young, 1991), which results in lower tail beat frequency (Wardle, 1975). Consequently, a loss in swimming speed and hydrodynamic lift is compensated for with proportionally wider caudal fin spans (Motani, 2002; Lingham-Soliar, 2005).”
L233 – Suggest “where hydrodynamic lift is not as important, with decreased need to control …”
L237 – Confusing sentence, suggest “Our work is the first to establish a comparative framework with geometric morphometrics of body-caudal fin propulsion on which to base ecomorphological inferences of extinct early vertebrates.”
L240 – Suggest “… approach is applicable for …”
L245 – Suggest “… placoderms and ‘acanthodians’)…”
L247 – Should be “… that contribute to buoyancy … ”
L253 – Suggest “ … remains of large pelagic arthrodires… “
L254 – Suggest “… make it difficult to base…”
L258 – Unclear, I suggest “As in modern sharks, arthrodire…”
L260 – Suggest “ … as the main propulsor organ, are also be subject to strong selection pressures driven by ecology, locomotion and body size”
L266 – ‘In this sense’ is not the best way to phrase this, and th term cruising describes the contuous swimming. Suggest “Recently, Carr (2010) has provided strong statistical evidence for active pelagic cruising in this species, based on taphonomical and sedimentological data.”
L270 – Autocorrect has resulted in typos here “… Basin (EEUU) and the Tafilalt Basin (Morocco), suggesting a…”
L276 – Suggest “… modes could also be present …”
L287 – Suggest “… armoured animals, for example arthropods… “
L293 – Suggest “ … thus constituting direct evidence of the …”
L307 – Misuse of the word funded again, suggest “…placoderms have been based on the proportions of …”
L314 – To clarify I suggest “ …, including C. cuspidatus itself (Miles & …”
L318 – Typo, should be “ variables in placoderms (see Carr….”
L323 – I would try and reduce the length of this sentence.
L343 – This sentence is unclear, what outlines? I think this needs to be more specific.
L346 – This sentence does not provide additional information and repeats previous points. I would suggest deleting it. I am also unclear as to what is being suggested in the line 348 sentence starting “In this sense…”. Again not a common phrase in scientific writing.
L352 – Need to explain the significance of the single dorsal fin and Coccosteus here. Sentence is unclear.
L357 – Need to define macruriform
L374 – Should be “ … kind of approach can be …”
L376 – Should be “ … in order to make inferences about …”
L379 – Should be “…sensitive to convergence…”
L384 – Should be “ … us to know what some of the first vertebrates…”. However, placoderms are far far from the first vertebrates, so I would reconsider this sentence.
L388 – The closing remarks are a little long and repetitive, I would consider rewriting these.

• Intro & background to show context. Literature well referenced & relevant.
L66 – would be beneficial to include a date with reference here
L76 – again, include a date and reference here
L291 – explain what Orodus is, a smaller fish?
Aside from these minor points, I think more emphasis needs to be placed on the controls on morphology. Function is mentioned but not discussed in detail. The relationship between jaw size and ecology again is not discussed. So I feel that there needs to be a more comprehensive introduction to the theory. E.g. why do fast cruising sharks have the tails they do?


• Structure conforms to PeerJ standards, discipline norm, or improved for clarity.
Yes. But please see notes above.
• Figures are relevant, high quality, well labelled & described.
Yes.

Experimental design

• Original primary research within Scope of the journal.
This article provides a good framework for the estimation of Dunkleosteus size; a large and iconic fish. However, the sole comparison with the great white jaws is not justified adequately in text.
• Research question well defined, relevant & meaningful. It is stated how the research fills an identified knowledge gap.
The research question is well-defined and relevant for the estimation of fossil fish body size. However ecological correlates are neglected, which are an important part of the study.

• Rigorous investigation performed to a high technical & ethical standard.
In essence this is a statistical approach to reconstructing Dunkleosteus postcrania. While competent, there is little ecological justification presented to support the findings, and I feel a more the problem deserves a more thorough approach.

• Methods described with sufficient detail & information to replicate.
L145 – Might be beneficial to explain what the pinnochio effect is here or in the introduction.
L149 – unclear as to why you are using 33 white sharks, and not all shark species here. Is there evidence for a direct comparison with a single species?
L205 – Should be “… structure involved in thrust generation …”

• Raw data supplied (see PeerJ policy).
Yes. Plots in Supplementary data 3 may need axis labels or for clarity. Crosses and circles need to be explained.

Validity of the findings

Impact and novelty not assessed. Negative/inconclusive results accepted. Meaningful replication encouraged where rationale & benefit to literature is clearly stated.

• Data is robust, statistically sound, & controlled.
While an impressive array of statistical tests have been utilised, I do not believe enough justification has been provided for the comparison of Dunkleosteus with the great white shark. This justification is a very important part of the analysis, since the authors themselves state the variability of morphology based on ecological and functional factors. For the comparison to be valid, I would need to see more evidence that the feeding mode and locomotion style of the two were very similar. The alternative is to run a similar analysis of other sharks of similar size (e.g. whale sharks, basking sharks, Greenland shark etc) to show a universal trend separated from ecology.
Secondly, the predictive power of this technique is stated on numerous occasions (e.g. L323); however the authors have not applied it to modern sharks to test this. E.g. Does the regression predict the size of a whale shark based on jaws?

• Conclusions are well stated, linked to original research question & limited to supporting results.
L268 – Needs a reference
Otherwise the links to original research are good.

Conclusion is a little repetitive, which reflects the limited scope of the investigation. This is not necessarily a bad thing.

• Speculation is welcome, but should be identified as such.
Speculation is separated from fact well.

Additional comments

This is a competent statistical comparison of Dunkleosteus with modern sharks. The question is not novel, but the approach used here is for this species.

I like this paper a lot, and it deserving of publication, but not in its current form. The comparison of jaws with only the great white shark is not justified or explained in text. The regression used to predict body size or tail type in an extinct animal is not tested here on modern sharks. Were the authors to choose a large pelagic plankton-feeding shark and predict tail span, this may go some way to testing the validity of their 'comparative framework'.

Reviewer 3 ·

Basic reporting

This is a really clever study which seeks to reconstruct the unknown body shape and size of a large, extinction charismatic megafauna (Dunkleosteus) using inference from living analogues among the sharks through the inference of general rules for fish body form. However, this worthy effort has a number of important issues which reduce the strength of the arguments and the ability to infer the morphology of extinct fishes with high probability; These need to be resolved before publication. It requires better organization, stronger justification of specific assumptions and arguments, and reconsideration of the methods.

Experimental design

Comparison of sharks and placoderms: I can see the basic logic for comparing Dunkleosteus exclusively with living sharks. Dunkleosteus is a large, generalist predator with sharp-edged jaws found in an open water setting, so modern great whites and other sharks are likely the best modern analogues. However, this argument is not explicit in the paper, nor is the case for selecting actively pelagic sharks as the group with the right body shape and proportions made clear. There needs to be a real comparison of the head and jaws and head size of Dunkleosteus and sharks to determine which shark ecological group is the ideal analogue. Reliance on past speculation is not enough.
Shark morphospace analysis and results: The results of the morphospace analysis are a bit muddled - there's quite a lot of overlap between sharks with different ecologies near the center of PC1 and along most of PC2. How do we know Dunkleosteus exhibited a morphology matching that of active pelagic sharks at their most distinct (e.g. the 4 with the most negative scores on PC1) and not at their most generalist (the other 6 points)? It is possible the restricted set of landmarks artificially increased the degree of overlap between different shark ecological groups. Why was body depth/shape and first dorsal fin position/form not captured by the landmark set? These aspects also vary in sharks, are important aspects of fish form related to locomotion and ecology, and are unknown in Dunkleosteus. Indeed, the body depth of the Dunkleosteus reconstruction in Fig. 4B is much greater than that of the sharks used as as analogues for inferring its body size and caudal shape, and the dorsal fin is unlike that of known large, pelagic fishes. It would be better to base all aspects of the body on quantitative inference. It is also not clear whether caudal fin edge shape, another ecologically relevant trait, was recorded, as the landmark sets in Fig. 1A and 1B disagree.
Dunkleosteus life-mode: This paper also needs more explicit justification for why Dunkleosteus was likely to be an active pelagic, "body and caudal fin"-type continuous swimmer than just citing Denison and Carr, and more shark-like than teleost-like. Some of the required evidence is referenced throughout the discussion (e.g. sedimentological evidence; the paragraph from 283-302 which should be moved forward) but not elucidated up front. A large range might be evidence, but body and caudal fin swimming, and continuous swimming, are not required for such (see the spread of Indo-Pacific reef fishes, for example).
Inference of body length: It is not clear why body length was estimated only from great white sharks. Are body length/jaw length ratios already known to be more dependent on ecomorphology across sharks, or fishes, than ancestry? While an argument is made for not estimating size based on smaller arthrodires, would it not be more appropriate to apply the shark scaling formula to placoderm relationships than assume the same relationship as in one species of shark?

Validity of the findings

Dunkleosteus reconstruction: While it is likely that Dunkleosteus did have a caudal form resembling that of modern pelagic sharks given what we know about fish ecomorphology, the arguments in the paper are not yet strong enough to support this assumption as laid out above. In addition, the new reconstruction of Dunkleosteus may have a correct tail, but the body depth, and other fin forms are still based on Coccosteus, which the authors clearly argue is not a good analogue. A full attempt to infer the form of early vertebrates requires consideration of these ecologically important aspects as well. An improved landmark set for sharks would go a long way to resolving this.

Additional comments

line 72: "pteraspidimorphs"

193: "hypocercal"

269: Appalachian Basin, Tafilalt Basin

280: thunniform

285: Lamsdell and Braddy 2009 did not explicitly test the trophic position of Dunkleosteus.

360: This section seems unnecessary. Paleoart should be based on the most up to date reconstructions as a matter of course and good paleoartists know this.

Fig. 1 The landmark sets in Fig. 1A and 1B disagree.

Fig. 4A It is not clear from the paper where these outlines come from.

Fig. 4B The reconstruction Dunkleosteus has much greater body depth than the sharks which provided the inferred caudal fin shape. Why is this assumed? Also, why are the median and paired fins assumed to be more like Coccosteus than pelagic sharks, since the shapes of these are also related to locomotion?

---

## Round 0.2 · accepted · Accept

Dear Dr Ferron,

I am pleased to inform you that your manuscript "Ecomorphological inferences in early vertebrates: reconstructing Dunkleosteus terrelli (Arthrodira, Placodermi) from palaeoecological data", co-authored with Martínez-Perez and Botella, is now accepted for publication in PeerJ.

Thank you again for consider PeerJ and we look forward to your future contributions to the Journal.

cheers,

Claudia Marsicano